# Chemical Characteristics of Ethanol and Water Extracts of Black Alder (*Alnus glutinosa* L.) Acorns and Their Antibacterial, Anti-Fungal and Antitumor Properties

**DOI:** 10.3390/molecules27092804

**Published:** 2022-04-28

**Authors:** Agnieszka Nawirska-Olszańska, Ewa Zaczyńska, Anna Czarny, Joanna Kolniak-Ostek

**Affiliations:** 1Department of Fruit, Vegetable and Plant Nutraceutical Technology, Wrocław University of Environmental and Life Sciences, 37 Chelmonskiego Street, 51-630 Wroclaw, Poland; agnieszka.nawirska-olszanska@upwr.edu.pl; 2Hirszfeld Institute of Immunology and Experimental Therapy, Polish Academy of Sciences, 12 R. Weigla Street, 53-114 Wroclaw, Poland; ewa.zaczynska@hirszfeld.pl (E.Z.); czarnyanna7@gmail.com (A.C.)

**Keywords:** UPLC-MS, biological properties, anti-cancer effect, polyphenolic compounds, extraction

## Abstract

The aim of this study was to identify polyphenolic compounds contained in ethanol and water extracts of black alder (*Alnus glutinosa* L.) acorns and evaluate their anti-cancer and antimicrobial effects. The significant anti-cancer potential on the human skin epidermoid carcinoma cell line A431 and the human epithelial cell line A549 derived from lung carcinoma tissue was observed. Aqueous and ethanolic extracts of alder acorns inhibited the growth of mainly Gram-positive microorganisms (*Staphylococcus aureus*, *Bacillus subtilis*, *Streptococcus mutans*) and yeast-like fungi (*Candida albicans*, *Candida glabrata*), as well as Gram-negative (*Escherichia coli*, *Citrobacter freundii*, *Proteus mirabilis*, *Pseudomonas aeruginosa*) strains. The identification of polyphenols was carried out using an ACQUITY UPLC-PDA-MS system. The extracts were composed of 29 compounds belonging to phenolic acids, flavonols, ellagitannins and ellagic acid derivatives. Ellagitannins were identified as the predominant phenolics in ethanol and aqueous extract (2171.90 and 1593.13 mg/100 g DM, respectively) The results may explain the use of *A. glutinosa* extracts in folk medicine.

## 1. Introduction

The alder (*Alnus* Mill.) is a genus of trees and shrubs from the birch family (*Betulaceae A. Gray*). They occur naturally in the temperate and boreal zones of the northern hemisphere. Black alder (*Alnus glutinosa* L.) acorns are rich in bioactive ingredients. Many studies have revealed that the extracts obtained from *Alnus* leaves, acorns and bark have antioxidant, anticancer, anti-inflammatory, antimicrobial and antiviral effects. These properties depend on the effects of polyphenols, terpenoids, steroids and numerous other compounds [1,2]. Abedini et al. [3] studied the activity of a crude methanol extract from the bark of *A. glutinosa* against bacteria and yeast. They found that oregonin was the most active growth-inhibiting compound against *Staphylococcus aureus*. Dahija et al. [2] found that methanolic extracts from alder leaves and bark strongly inhibited the growth of the Gram-positive bacteria, *Staphylococcus aureus* and *Bacillus subtilis*. These extracts contained high levels of phenols or flavonoids. Li et al. [4] identified betulin, betulinic acid and betulone as the major anti-mycobacterial substances in the bark of *A. incana*.

To take advantage of the benefits of such compounds in plants, researchers must use appropriate extraction techniques. The most important step in determining polyphenol compounds is the extraction process and identification of bioactive compounds [5]. However, the range of structures for polyphenols is so wide that creating a standard extraction protocol to extract all target compounds from different plant materials is impossible [6]. Previously published results have shown that mixtures of alcohols and water are more effective in extracting phenolics compared to mono-component solvent systems [7]. The reason is that water is able to increase the diffusion of polyphenols from plant cells through the swelling effect of plant tissues [8].

Traditionally, extracts of the Alnus species are used for gastrointestinal and skin diseases as well as bacterial infections of the mouth and throat [2]. Natural products derived from plants are used to treat cancer due to fewer side effects compared to the standard treatments available for cancer [9,10]. Several plant-derived extracts and molecules inhibit and regulate the signaling processes and networks associated with the growth and proliferation of cancer cells [11,12]. Recent studies have shown that medicinal plant-derived extracts can regulate the apoptosis of cancer cells [13]. Extracts of some Alnus species have immunostimulatory properties and are used in anticancer therapy [14].

Bacterial infection lesions still constitute a serious problem, because microorganisms are resistant to numerous antibiotics and disinfectants. *Staphylococcus aureus*, *E. coli* and *Pseudomonas aeruginosa* are important opportunistic pathogens, commonly found as part of the normal microflora of different parts of the body, without causing diseases in healthy humans. In people with low immunity, *S. aureus* and *P. aeruginosa* infections are usually related to soft tissues and skin and can cause very serious diseases in these individuals [15].

Plant secondary metabolites are suitable in this respect because they can enhance the action of antibiotics as well as affecting the body’s immunity. Many plants’ bioactive compounds have been investigated against pathogenic bacteria, especially Gram-positive and Gram-negative and fungi. One large group of naturally occurring bioactive substances comprises flavonoids and polyphenols. These substances influence bacterial membrane permeability and nucleic synthesis, can reduce virulence, neutralize bacterial toxins, hinder biofilm formation and inhibit the entry of the host cells into the cytosol [16,17]. Determining the composition and antimicrobial activity of acorn extracts could open up new possibilities for fighting bacterial infections.

The aim of our research was to identify the chemical components of water and ethanol extracts from black alder (*A. glutinosa* L.) acorns and determine their biological properties. There are no published data on the anticancer and antimicrobial activities of water or ethanol extracts from black alder (*A. glutinosa* L.) that naturally grow in Kalisz, Poland. Additionally, there are no published data on the chemical characterization of extracts from the black alder acorns. The presented results indicate for the first time the possibility of using this organ of the alder as a donor of bioactive substances with a pro-health effect.

## 2. Results and Discussion

### 2.1. Identification of Phenolic Compounds in Ethanol and Aqueous Extracts of A. Glutinosa Acorns

Table 1 and Figure 1 list the 29 compounds identified in ethanol and aqueous extracts of black alder acorns, through UPLC-MS/MS (with PDA and Q/TOF detectors) experiments, along with their retention times (Rt) and UV-vis spectral profiles at 200–600 nm. Molecules that were certainly or putatively identified in negative ion mode belong to the compound groups of phenolic acids, flavonols, ellagitannins and ellagic acid derivatives. Twenty-two compounds—fifteen ellagitannins (**1**, **3**–**11**, **14**–**16**, **19** and **21**), two phenolic acids (peaks **2** and **12**), three ellagic acid derivatives (**23**, **24** and **28**) and two flavonols (**27** and **29**)—were detected in the aqueous extract. In the ethanolic extract **17** compounds—thirteen ellagitannins (peaks **10**, **13**–**22**, **25** and **26**), two ellagic acid derivatives (**23** and **24**) and two flavonols (**27** and **29**)—were detected.

#### 2.1.1. Ellagitannins

High MW ellagitannins based on galloyl-bis-HHDP-glucose units, such as pedunculagin [18] and castalagin, were already found in the Alder species. Compounds **1** and **3** were identified as HHDP-hexosides (Table 1) based on their (M − H)^−^ ion at *m*/*z* 481.06 and on the resulting product ions at *m*/*z* 300.99, consistent with the data reported in the literature for this compound [19]. Compound **4** had (M − H)^−^ at *m*/*z* 649.05 and was identified as trigalloyl-glucose. The MS/MS fragmentation gave an ion with (M − H)^−^ at *m*/*z* 605.07, which corresponds to the loss of CO_2_ (44 Da), a fragment with *m*/*z* at 479.05 owing to the loss of a galloyl moiety (152 Da) and a fragment with *m*/*z* at 300.99, corresponding to ellagic acid, in agreement with the literature data for this compound [19]. The TOF-MS analysis of compound **5** revealed a deprotonated molecule at *m*/*z* 963.13. The MS/MS fragmentation gave an ion with (M − H)^−^ at *m*/*z* at 933.06 corresponding to the polyphenol castalagin or its isomer vescalagin and a fragment with *m*/*z* at 300.9999, corresponding to ellagic acid. This compound was suggested as methoxylated castalagin/vescalagin, after data published by Sobeh et al. [20]. Peaks **6** and **15** have an (M − H)^−^ at *m*/*z* 785.0615 and were identified as HHDP-digalloyl-hexoside isomers. The MS/MS fragmentation gave an ion with (M − H)^−^ at *m*/*z* at 633.06, owing to the loss of a galloyl moiety (152 Da) and a fragment with *m*/*z* at 300.9999, corresponding to ellagic acid [21,22]. Compounds **7**, **9**, **10**, **13** and **19** were identified as pedunculagin (bis-HHDP-hexoside) isomers (Table 1) based on their (M − H)^−^ ion at *m*/*z* 783.07 and on the resulting product ions at *m*/*z* 481.06 and 300.99, which is consistent with the data reported in the literature for this compound [19]. The TOF-MS analysis of compound **8** revealed a deprotonated molecule at *m*/*z* 483.07 corresponding to digalloyl-hexoside [22]. Peaks **11** and **16** were identified as HHDP-galloyl-hexosides, based on their (M − H)^−^ ion at *m*/*z* 633.0550 and on the resulting product ions at *m*/*z* 481.04 and 300.99, corresponding to the loss of a galloyl moiety (152 Da) and ellagic acid. The TOF-MS analysis of compounds **14**, **18**, **25** and **26** showed deprotonated molecules at *m*/*z* 935. The MS/MS fragmentation gave an ion with (M − H)^−^ at *m*/*z* at 783.06 corresponding to the loss of a galloyl (152 Da) moiety, a fragment with *m*/*z* at 633.07, corresponding to the galloyl-HHDP-glucose structure and a fragment with *m*/*z* at 300.99, corresponding to ellagic acid. These compounds were suggested to be galloyl-bis-HHDP-hexose isomers, after data published by Carocho et al. [23]. The ESI–MS spectrum of compounds **17**, **20** and **22** showed a deprotonated molecule at *m*/*z* 933. The MS/MS spectrum of the deprotonated molecule produced fragments at *m*/*z* 915, 633, 450 and 300.99, which are in agreement with those attributed to castalagin or vescalagin isomers [23]. The TOF-MS analysis revealed the presence of sanguiin H-6, a dimeric ellagitannin (peak **21**, *m*/*z* 1869.14). The MS/MS fragmentation gave an ion with (M − H)^−^ at *m*/*z* at 935, owing to the loss of galloyl-bis-HHDP-glucose (935 Da), a fragment with *m*/*z* at 633, corresponding to the galloyl-HHDP-glucose structure, a fragment with *m*/*z* at 468.99, corresponding to sanguisorbic acid dilactone and a fragment with *m*/*z* at 300.99, corresponding to ellagic acid. This compound was identified in alder acorns for the first time.

#### 2.1.2. Ellagic Acid Derivatives

The examination of the UPLC chromatograms in TOF-MS mode revealed the presence of free ellagic acid (peak **24**, *m*/*z* 300.9999) (Table 1). MS/MS fragments at *m*/*z* 285.04, 257.02 and 229.01 are characteristic for ellagic acid, and are consistent with the data reported in the literature for this compound [24]. Peak **23** has a pseudomolecular ion at *m*/*z* 433.04 and an MS/MS fragment at *m*/*z* 300.99, owing to the loss of a pentoside residue (132 Da), while peak **28** has a pseudomolecular ion at *m*/*z* 461.01 and an MS/MS fragment at *m*/*z* 300.99, owing to the loss of a hexoside residue (162 Da). These compounds were identified as ellagic acid pentoside and hexoside, respectively.

#### 2.1.3. Phenolic Acids

The TOF-MS analysis revealed the presence of two phenolic acids in the investigated aqueous extracts of alder acorns, using negative ionization mode (Table 1). Compound **2** with (M − H)^−^ at *m*/*z* 355.03 was characterized as ferulic acid hexoside. The MS/MS fragment at *m*/*z* 193.01 is characteristic of ferulic acid, and corresponds to the loss of a hexoside residue (162 Da). The TOF-MS analysis of compound **12** revealed a deprotonated molecule at *m*/*z* 367.00. The MS/MS fragmentation gave an ion with (M − H)^−^ at *m*/*z* at 191.01 corresponding to quinic acid. This compound was suggested as methyl-caffeoyl-quinate, after data published by Fernández-Poyatos et al. [25].

#### 2.1.4. Flavonols

The examination of the chromatograms in TOF-MS mode revealed the presence of two flavonols in the examined alder acorns, using negative ionization mode (Table 1). Peak **27** (*m*/*z* 447.05) was characterized as isorhamnetin pentoside on the basis of its fragmentation pattern, with a loss of pentose residue (132 Da). Peak **29** had a pseudo-molecular ion at *m*/*z* 315.01 and was identified as isorhamnetin, based on a comparison with a standard reference compound.

### 2.2. Quantitative Analysis of Polyphenols

The concentration range of total phenolics determined in different extracts of alder acorns is presented in Table 2. The quantification of phenolic acids, flavonols, ellagitannins and ellagic acid derivatives was performed using authentic standards.

The ethanol extract of alder acorns was characterized by a higher concentration of phenolic substances (3276.72 mg/100 g DM) than in the aqueous extract, which contained 2849.93 mg of phenolic compounds in 100 g of dry matter. Ellagitannins were identified as the predominant phenolics in ethanol and aqueous extract (2171.90 and 1593.13 mg/100 g DM, respectively). In the aqueous extract no castalagin/vescalagin presence was found, while in the ethanol extract the amount of these compounds was 501.60 mg/100 g DM. In addition, galloyl-bis-HHDP-hexose was found only in the ethanolic extract, and its amount was 448.47 mg/100 g DW. HHDP-hexosides, trigalloyl hexose and digalloyl hexose were found only in the aqueous extract, and their amounts were 186.78, 68.04 and 82.31 mg/100 g DM, respectively (Table 2). Ellagic acid derivatives were found in both aqueous and ethanolic extracts, but in the ethanol extract their concentration was higher, at 992.54 mg/100 g DM. Ellagic acid hexoside (29.13 mg/100 g DM) was found only in the aqueous extract. Phenolic acids were found only in the aqueous extract (263.57 mg/100 g DM). The amount of flavonols was three times higher in the ethanol extract, at 112.28 mg/100 g DM (Table 2).

The type of extraction solvent as well as the extraction procedures may have a significant impact on the profile of extracted polyphenols from plant material. It is related to the differences in the phenolic compounds’ structure, which determine their polarity and their solubility in different solvents [26].

The present results, showing a higher content of polyphenols in ethanol extracts, are similar to the results obtained by other researchers. Koffi et al. [27], in their research of twenty-three Ivorian plants, reported a four times higher amount of total phenolic content in ethanol extracts than in aqueous, acetone or methanol extracts. On the other hand, Onyebuchi and Kavaz [28], in their study on *Ocimum gratissimum* L. plants, obtained different results. Aqueous extracts obtained in different temperatures were characterized by a higher total phenolic concentration than ethanol extracts. In addition, Zeroual et al. [29,30], in their study on wild Chamomile and Rosemary, observed that extraction with methanol showed the highest yield, total phenolic and total flavonoids content, regardless of the extraction method used. Additionally, Kobus et al. [31] stated that not only type but also concentration of the used solvent has a significant impact on extraction efficiency. In their study the highest extraction yield was obtained with a mixture of water and ethanol (60% ethanol) and the lowest with ethanol (96%). The use of pure water as a solvent produced about 11% higher yield than 96% ethanol. Arize et al. [32] found in their research that water extracts more phenolic compounds from a solid than ethanol, due to a higher polarity factor. Due to a different molecular structure, ethanol may more effectively wash out compounds such as flavonoids [33]. Therefore, a mixture of water and ethanol (3:7, *v*/*v*) will provide a higher extraction efficiency for polyphenolic compounds than the separate use of these solvents.

Differences in the concentration of phenolic compounds in the tested extracts may also be a consequence of the extraction temperature. High water temperature during extraction causes cracking of the plant cell walls, which in turn leads to diffusion of plant components into the aquatic environment. Water is a more effective solvent due to a higher polarity coefficient [34]. However, the lower concentration of polyphenolic compounds in the aqueous extract compared to the ethanol extract may be due to the reduced extraction conditions, as well as the solubility of the compounds in water. In addition, an increase in temperature causes a decrease in the dielectric constant of water, and thus fewer polar compounds will be dissolved in it [28].

The antioxidant activity of phenolic compounds is the chemical feature that has received the greatest attention. Furthermore, these substances’ antioxidant activity is frequently associated with antiviral and antibacterial activity [35,36].

For many years, scientists have been studying the relationship between the structure and antioxidant activity of phenolic chemicals. The results acquired thus far have allowed for the determination of general relationships, i.e., it has been demonstrated that the presence of free hydroxyl groups and their mutual position determines the antioxidant activity of a chemical [37]. Furthermore, investigations in various model systems have led to the identification of functional groups in flavonoid molecules that are responsible for the activity in the investigated system. Wang et al. [37] found that the 4’-OH group in the B-ring had the maximum reduction potential with regard to superoxide radicals, after studying the antioxidant activity of flavonoid aglycones such as fisetin, kaempferol, morin, myricetin and quercetin. Tocopherols, flavonoids and phenolic acids are the most important natural antioxidants. The phenolic hydroxyl group is the most quickly oxidized of the phenolic hydroxyl groups, with the ability to trap free radicals such as ROS and active nitrogen species [38,39].

### 2.3. Effect of the Tested Extracts on Growth of Tumor Cell Line

The human skin epidermoid carcinoma cell line (A431) and the human lung epithelial cell line (A549) were cultured along with varying dilutions of ethanol or black alder acorn extract (1:400–1:3200) for 72 h, followed by cell viability determination applying the MTT assay. As shown in Table 3, significant growth-inhibitory effects were observed for all tested dilutions of plant extracts as compared to the control cultures.

With the reduction of ethanol or water plant extract dilution, the percentages of living cells decreased proportionally, indicating that the extracts had a dose-dependent inhibitory effect on the tumor cells. Phase contrast microscopic observations of human skin carcinoma cells or a human lung cancer cell line after treatment with ethanol or water plant extracts for 72 h indicated low cell confluence. Furthermore, floating cells revealed that treatment with alder acorn extracts resulted in a decrease of adherence. Untreated cells showed higher cell confluence and adherence to the culture plates under similar conditions. The nature and the differential features observed in the cancer cells clearly defined the cellular morphological characteristics, which were typical of cell death (not shown). The results of this study showed that the aqueous alder acorn extract showed significant antitumor activity against the A431 skin cancer cell line.

Plants are considered to be an important source of anticancer drugs; therefore, various natural compounds from plants have been used as anticancer drugs. The present study investigated the anticancer activity of aqueous or ethanol alder acorn extract against the A431 skin and A549 lung cancer cell lines. Antiproliferative effects were investigated using an MTT assay, and activity was analyzed in a set of morphological features. The assay detects the reduction of MTT salt to the blue formazan product by mitochondrial dehydrogenase, which indicates cell viability. Vijaybabu and Punnagai [10] used this same MTT method to study the anti-proliferative properties of the composition of vanilla leaves against the skin cancer cell line A431 and the cytotoxic activity of the grape seed extract was also observed against A431 using MTT assay [40]. Other authors demonstrated that a Scutellaria barbata ethanol extract showed anti-tumor activity in vitro and could inhibit growth of the human lung cancer cell line, A549. The basic mechanism of inhibition was due to cell apoptosis and cytotoxic effects [41].

Polyphenols have received a lot of attention in cancer therapy because of their chemopreventive effects as both blocking and suppressing agents [42,43]. In terms of blocking actions, polyphenols can prevent carcinogen activation, prevent reactive carcinogens from connecting with key DNA locations and aid in the metabolic clearance of carcinogens. Furthermore, polyphenols have the ability to decrease or suppress oncogenesis and cancer progression, allowing them to exert chemopreventive effects at multiple phases of carcinogenesis [44]. Polyphenols’ anticancer actions are multi-targeted and include the activation of many pathways to trigger apoptosis in cancer cells. Furthermore, three major epigenetic changes (alterations in chromatin structure, DNA methylation and microRNA regulation) are implicated in tumor cells treated with biomass polyphenols [45]. Although several targets for the antitumor/anticancer actions of biomass polyphenols have been identified, the detailed processes by which polyphenols are capable of influencing the expression of these genes/miRNAs remain unknown. It would thus be interesting to choose single or high purity polyphenols with strong antitumor/anticancer properties from natural product resources to further research their interactions with antitumor variables.

In the present study, our results showed that an aqueous extract obtained from acorns of black alder inhibited the growth of tumor cells more effectively than ethanol extract. The differences in the effect of the tested extracts may result from the different amounts of phenolic compounds (phenolic acids, flavonols, ellagitannins and ellagic acid derivatives), and the extraction conditions (extraction temperature, solvent used or its concentration). The results of this study revealed that ethanol and water extracts of black alder (*A. glutinosa* L.) acorns displayed potent anti-proliferative activity against human cancer cell lines.

### 2.4. Antimicrobial Activity

The development of antibiotic resistance demands more effective antimicrobial therapy. Plants contain metabolites that exhibit antioxidant and antimicrobial properties. Polyphenols can support the action of antibiotics [46], therefore preliminary studies have been carried out on the effects of polyphenols from alder acorns on the growth of bacteria. Those microorganisms that are often antibiotic-resistant and are responsible for hospital infections that are difficult to eradicate were selected for the study. In the case of microbiological tests, a screening test was initially carried out in order to select the best extract. The antibacterial activity of water extract of alder acorns against Gram-negative (*Pseudomonas aeruginosa*, *Escherichia coli*, *Proteus mirabilis*, *Citrobacter freundii*) and Gram-positive (*Staphylococcus aureus*, *Streptococcus mutans*) strains as well as the antifungal activity against *Candida albicans* and *Candida glabrata* are presented in Table 4.

The aqueous extract of alder acorns inhibited the growth of mainly the Gram-positive microorganisms, *B. subtilis*, *S. aureus* and the yeast-like fungi *C. albicans* and *C. glabrata*. The S. aureus strain was found to be the most sensitive; the clear zone of growth inhibition was 12 mm. Of the Gram-negative bacteria, this extract inhibited the growth of only the *E. coli* strain. The activity of the extract depended on the extract dilutions; as can be seen from the table, the dilution of 1:10 was characterized by the weakest activity.

Subsequently, the antimicrobial effect of alcoholic solutions of these plants was examined. Ethanol solutions were initially diluted with water 20 times (alcohol concentration 3.5%). The ethanol extract from black alder acorns inhibited the growth of some strains. The results are illustrated in Table 4.

Due to the necessity of reducing the ethanol content to the control value which does not affect the growth of microorganisms, it is difficult to compare the activity of the two extracts. Based on the results of testing both extracts (for different dilutions), a higher sensitivity of the Gram-positive strain was found. Most of the Gram-negative strains tested were resistant; only *E. coli* was sensitive to this extract. The lower activity of the ethanol extract from alder acorns was due to the lower concentration of substances contained in it.

As shown in Table 4, the most sensitive to the action of Alnus acorns’ alcoholic extract were two Gram-positive strains, *S. aureus*, *Bacillus subtilis* and one Gram-negative strain *E. coli*. The ethanol extract had little effect on the strains of *C. albicans* and *C. glabrata*. Whereas Gram-negative bacilli, *Proteus mirabilis*, *Citrobacter freundii* and *Pseudomonas aeruginosa*, were resistant to the effect of both aqueous and ethanol extract from alder acorns, despite the differences in the composition of the extracts. The aqueous extract contained phenolic acids which were not found in the ethanol extract. In turn, the sum of phenolic compounds in the ethanol extract was higher (3276.72 mg/100 g DM) than in the aqueous extract (2849.93 mg/100 g DM). Gram-positive bacteria were more sensitive to polyphenols due to the structure of their cell wall. The antimicrobial activity of polyphenols depends on the reaction with the bacterial cell. The hydroxyl groups of phenolic compounds by binding with the proteins of bacterial cells inhibit the activity of enzymes, which affects the life reactions of bacteria.

The cell wall of Gram-negative bacteria is connected to the outer membrane, which limits the penetration of substances. In addition, some bacteria, such as P. aeruginosa, have developed a number of mechanisms that protect them against harmful substances. One of them is mucus secretion, which limits the contact of the cell wall with external factors. Another way is through biofilm formation or the operation of genetically determined multi-drug efflux pumps. Taguri et al. [47], in studies on the action of polyphenols on bacteria, observed that the antibacterial activity of these compounds largely depends on the species of microorganism and the structure of their cell wall.

Further research was aimed at determining the effect of the water extract on the number of bacteria. Two strains sensitive to this extract were selected, Gram-negative *E. coli* and Gram-positive *Staphylococcus aureus*.

The results of the antimicrobial activity of the water extract from acorns, determined by the plate method for quantifying live bacteria, are presented in Table 5.

The growth of *S. aureus* was markedly inhibited (10^5^) compared to the control (10^7^). In contrast, the number of *E. coli* bacteria decreased slightly (10^8^ to 10^7^). *S. aureus* is a microorganism widely distributed in the human population, and causes infections that are difficult to treat due to antibiotic resistance. Antimicrobial agents of plant origin can increase the activity of antibiotics and can be used as an alternative to drug development to combat resistant *S. aureus* infections. The diverse antibacterial and antifungal activity of the studied alder acorn extracts depended on the extraction method, chemical composition, concentration of these extracts as well as the type of microorganisms. The presented data show that Gram-positive microorganisms were more sensitive to the test extracts. Disk diffusion tests conducted by Dahija et al. [2] against Gram-negative, Gram-positive and fungal strains also showed that Gram-positive *Staphylococcus aureus* and *Bacillus subtilis* strains were more sensitive to alder and leaf bark extracts. This could be due to the presence of the outer membrane as a permeability barrier in Gram-negative bacteria. Altinyay et al. [48] determined the antibacterial effect of ethanolic and aqueous leaf extracts of Alnus glutinosa. No antibacterial activity was observed against *Bacillus subtilis*, *E. coli*, or *Pseudomonas aeruginosa* for the aqueous extracts. The ethanolic extracts exhibited antibacterial activity against all the tested strains. Our results are similar to those obtained by Altinyay et al. [48]. Water and ethanol extracts inhibited the growth of mainly Gram-positive bacteria, such as *Staphylococcus aureus*, *Candida albicans* and *Candida glabrata*.

Polyphenols are thought to be one of the most intriguing natural extracts for inhibiting bacterial growth and proliferation through a variety of mechanisms, including changes in bacterial membrane permeabilization, inhibition of bacterial DNA gyrase, interference with energy metabolism and disruption of bacterial porin functions. Furthermore, the presence of phenolic hydroxyl groups enhances polyphenols’ antibacterial effects by compromising the structural integrity and functionality of bacterial membranes [49,50,51,52]. The bacterial membrane is mostly made up of lipid bilayers with hydrophilic and hydrophobic ends. The binding of phenolic hydroxyl groups to hydrophilic ends causes membrane lipid aggregation, which destroys the bacterial membrane. Intriguingly, when polyphenolic compounds are inoculated in plants and fruits, they activate a couple of antibacterial enzymes, including phenylalanine aminase, catalase, peroxidase, polyphenol oxidase, chitinase and -1,3-glucanase, so boosting plant and fruit antibacterial powers [53]. Currently, extensive research on the antibacterial effects of biomass polyphenols is being conducted. However, greater research into the structure–function relationship of polyphenols, as well as the combinational uses of polyphenols with more evident antibacterial effects, is required.

## 3. Materials and Methods

### 3.1. Reagents and Standards

Acetonitrile was purchased from Merck (Darmstadt, Germany). Formic acid and methanol were purchased from Sigma–Aldrich (Steinheim, Germany). Ferulic acid, caffeic acid, isorhamnetin 3-*O*-glucoside and ellagic acid were purchased from Extrasynthese (Lyon, France). RPMI-1640 medium was purchased from Biowest (Nuaillé, France). Fetal bovine serum (FBS) was obtained from HyClone (Pittsburgh, PA, USA). L-glutamine, penicillin and streptomycin solution and MTT (3-(4,5-dimethylthiazol-2-yl)-2,5-diphenyltetrazoliumbromide) were purchased from Sigma-Aldrich (St. Louis, Mo, USA). Nutrient agar, brain heart broth, blood agar, Sabouraud broth, Sabouraud agar and 0.9% NaCl were purchased from BTL Sp. z o. o., Łódź, Department of Enzymes and Peptones (Staufen, Poland).

### 3.2. Plant Material

Black alder (*Alnus glutinosa* L.) acorns were used in this study. These were obtained from Kalisz, Poland. Acorns were picked up on the same day in early September, 2019. In the course of the studies, three replications (3 trees, 10 randomly chosen acorns, i.e., 30 replications) were carried out. After harvesting, acorns were air dried and crushed. The moisture content of the acorns was 11.17%. Homogeneous powders were obtained by crushing the dried tissues using a closed laboratory mill to avoid hydration (IKA 11A; Staufen, Germany). Powders were kept in a refrigerator (−80 °C) until analysis.

### 3.3. Extraction Procedure

In order to prepare the stock solution, 1 g of dried material was dissolved in 5 mL of 70% ethanol or H_2_O. The samples of plants were extracted using 70% ethanol via maceration in darkness at room temperature or being held at the boiling point for 60 min at 100 °C. After 24 h, extracts were centrifuged at 3600 rpm for 20 min and the supernatants were collected and stored at 4 °C until the testing.

### 3.4. Determination of Phenolic Compounds

#### Identification and Quantification of Phenolic Compounds by the UPLC-PDA–MS Method

The stock solutions were centrifuged at 19,000× *g* for 10 min, and the supernatant was filtered through a hydrophilic PTFE 0.20 µm membrane (Millex Simplicity Filter, Merck, Darmstadt, Germany) and used for analysis. Identification of polyphenols of *A. glutinosa* L. extracts was carried out using an ACQUITY UPLC system equipped with a PDA detector (Waters Corporation, Milford, MA, USA) and G2 Q-TOF micro mass spectrometer (Waters, Manchester, UK) equipped with an electrospray ionization source operating in negative mode. Identification and quantification of phenolic compounds were performed according to the method described by Kolniak-Ostek and Oszmiański [54]. The compounds were monitored at 254 nm (ellagitannins and ellagic acid derivatives), 320 nm (phenolic acids) and 360 nm (flavonols). Calibration curves were determined experimentally for caffeic, ferulic and ellagic acid and isorhamnetin 3-*O*-glucoside. Ferulic acid derivatives were expressed as ferulic acid, methyl-caffeoyl-quinate was expressed as caffeic acid, isorhamnetin derivatives were expressed as isorhamnetin 3-*O*-glucoside and ellagitannins and ellagic acid derivatives were expressed as ellagic acid All experiments were performed in triplicate. The results are expressed as milligrams per 100 g of dry matter (DM).

### 3.5. Determination of Biological Properties

#### 3.5.1. Cell Culture

A549 cells (ATCC CCL 185)—human epithelial cell line derived from lung carcinoma tissue–were obtained from the American Type Culture Collection (Manassas, VA, USA). A431 cells (ATCC CRL 1555)—human skin epidermoid carcinoma cell line–were obtained from the American Type Culture Collection (Manassas, VA, USA). The cells were cultured in RPMI-1640 and supplemented with 10% FBS, 100 units/mL penicillin and 100 µg/mL streptomycin in a humidified 5% CO_2_ atmosphere at 37 °C.

#### 3.5.2. Bacterial Strains

The experiments were carried out on strains from the Polish Collection of Microorganisms (PCM): *Pseudomonas aeruginosa* PCM 2058, *Pseudomonas aeruginosa* PCM 499, *Escherichia coli* PCM 2057, *Proteus mirabilis* PCM 2958, *Citrobacter freundii* PCM 2959, *Staphylococcus aureus* PCM 2054, *Bacillus subtilis* PCM 2021, *Candida glabrata* PCM 2703 and from the American Type Culture Collection (*Streptococcus mutans* ATCC 25175, *Candida albicans* ATCC 90028).

#### 3.5.3. Growth Inhibition of Tumor Cell Lines

The cells were re-suspended in the culture medium and distributed into 96-well flat-bottom plates. A-549 and A-431 were present at 2.5 × 10^4^ cells/well. The extract stock solution was tested at a ratio of 1:200 to 1:6400 (i.e., the final part of the extract in the culture medias). After 3-day incubation in a cell culture incubator, proliferation was determined using the MTT colorimetric method. In parallel, control cultures containing appropriate dilutions of the solvent (H_2_O or ethanol) were also incubated. The results were presented as mean optical density (OD) values from three wells ± standard deviation.

#### 3.5.4. Colorimetric MTT Assay for Cell Growth and Viability

Briefly, 25 µL of MTT (5 mg/mL) stock solution was added per well at the end of the cell incubation period and the plates were incubated for an additional 3 h in a cell culture incubator. Then, 100 µL of the extraction buffer (20% SDS with 50% DMF, pH 4.7) was added. After overnight incubation the optical density was measured at 550 nm with the reference wavelength of 630 nm in a Dynatech 5000 spectrophotometer [55].

#### 3.5.5. Determination of Antimicrobial Activity

The antimicrobial activity of the preparations was determined by the microorganism growth inhibition assay. The following culture media were used: nutrient broth, nutrient agar, brain heart broth, blood agar, Sabouraud broth, Sabouraud agar and 0.9% NaCl.

The pre-cultures of *P. aeruginosa*, *E. coli*, *P. mirabilis*, *C. freundii*, *B. subtilis* and *S. aureus* in nutrient broth and the pre-cultures of *S. mutans* in brain heart broth were incubated at 37 °C. The pre-cultures of *C. albicans* and *C. glabrata* in Sabouraud broth were incubated at 28 °C. After an overnight incubation the cultures were 10× diluted with broth. The suspensions of the microorganisms (100 µL) were seeded into plates with nutrient agar, blood agar or Sabouraud’s agar. After the drying of plates, 20 µL of the studied water or alcohol plant extracts at the indicated dilutions were spotted. When the sample showed antimicrobial activity, after 24 h, a clear growth inhibitory or opaque zone was observed on the agar.

In further studies, the plate method for quantifying viable bacteria was used. The two strains Gram-positive *Staphylococcus aureus* and Gram-negative *E. coli* were chosen. Pre-cultures in the culture broth were incubated for 18–20 h at 37 °C. Into the wells of 24-well culture plates 1 mL of the bacterial culture was applied, followed by the addition of the water extract of alder acorns (5× dilution). After 24 h incubation 100 µL of the bacteria culture was aspirated, diluted with 0.9% NaCl and seeded (100 µL) onto agar plates. After overnight incubation at 37 °C, the numbers of colonies (colony forming units—CFU) were counted. Bacteria incubated without acorn aqueous extract were the control.

#### 3.5.6. Statistics

All experiments were performed in triplicate and repeated at least three times. The data are presented as mean values ± standard deviation (SD). The Brown–Forsythe test was used to determine the homogeneity of variance between groups. When the variance was homogenous, analysis of variance (One-way ANOVA) was applied, followed by post hoc comparisons with Tukey’s test to estimate the significance of the difference between groups. Statistical significance level was determined at *p* < 0.05. The statistical analysis was performed using STATISTICA 7.0 for Windows.

## 4. Conclusions

Black alder (*A. glutinosa* L.) ethanolic and aqueous extracts show a wide spectrum of phenolics belonging to the group of phenolic acids, flavonols, ellagitannins and ellagic acid derivatives. The high antitumor and antibacterial activity in vitro is related to the high content of polyphenols and the activity of antioxidants. Our preliminary research shows that the new alder acorns’ polyphenol extracts have antibacterial properties. This activity depended not only on the concentration and type of polyphenols in the extracts, but also on the microorganism strains. Despite the differences in the amount and composition of polyphenols between the aqueous and ethanolic extract, the antimicrobial and antitumor activity was similar. Among the examined microorganisms, Gram-positive ones were more sensitive than Gram-negative. This suggests that *A. glutinosa* extracts have potential applications in the prevention of non-infectious diet-related diseases such as cancer and chronic inflammation. Moreover, they can be a safe alternative to bactericides, viruses and fungicides currently used in industry. In conclusion, to the best of our knowledge, this is the first study to isolate and to examine water and ethanol extracts obtained from the acorns of black alder (*A. glutinosa* L.) for their anti-cancer and anti-bacterial activity.

## Figures and Tables

**Figure 1 molecules-27-02804-f001:**
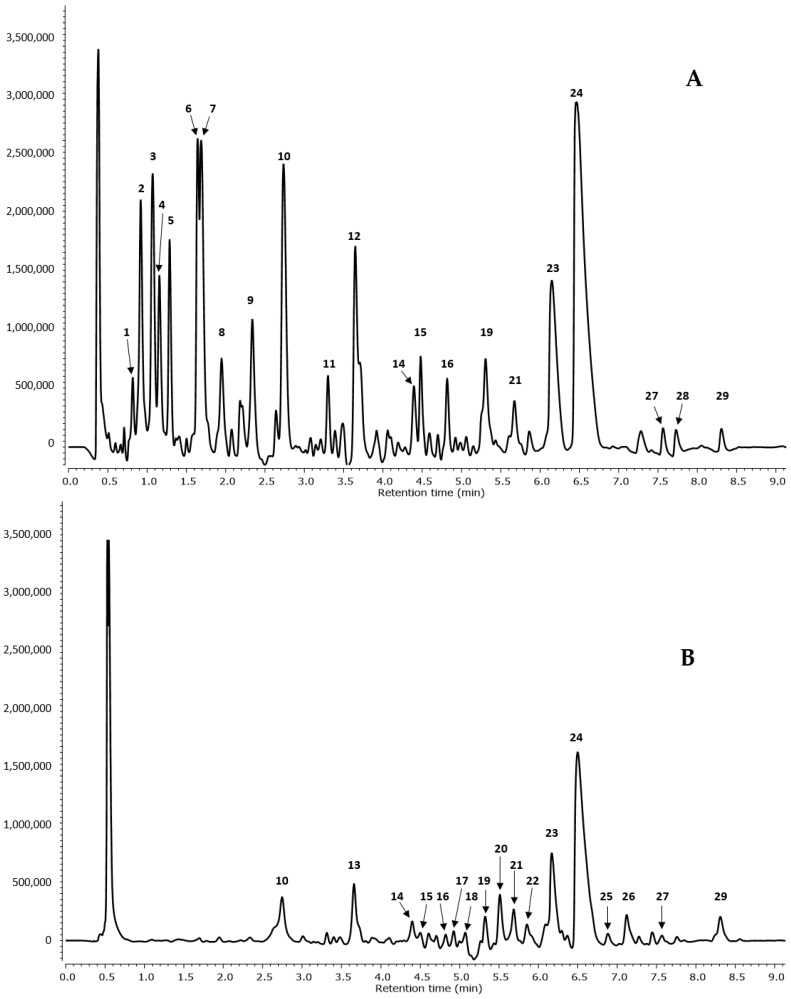
UPLC–MS chromatogram profile of *Alder* acorns (**A**) water and (**B**) ethanol extracts at 280 nm. Peak number identities are displayed in Table 1.

**Table 1 molecules-27-02804-t001:** Identification of phenolic compounds in *Alder* acorns ^a^.

Peak	Rt	λ (nm)	[M − H]^−^ (*m*/*z*) ^b^	MS/MS (*m*/*z*) ^b^>	Tentative Identification	Water Extract	Ethanol Extract
1.	0.82	220	481.0628	300.9964/275.0187/257.0077/229.0715	HHDP-hexoside(1-galloyl-2,3-HHDPl-α-glucose)	√	
2.	0.92	324	355.0305	193.0130	Ferulic acid hexoside	√	
3.	1.07	222	481.0611	300.9986	HHDP-hexoside	√	
4.	1.16	273	649.0592	605.0702/479.0501/300.9987	Trisgalloyl hexoside	√	
5.	1.29	230	963.1341	933.0640/300.9985	Methoxylated castalagin/vescalagin	√	
6.	1.64	235/320	785.0615	633.0641/300.9984	HHDP-digalloyl-hexoside	√	
7.	1.71	274	783.0668	481.0624/300.9978	Bis-HHDP-hexoside (pedunculagin isomer)	√	
8.	1.95	280	483.0761	271.0187/193.0340/169.0134/125.0235	Digalloyl-glucose	√	
9.	2.34	274	783.0645	633.0770/481.0583/300.9964	Bis-HHDP-hexoside (pedunculagin isomer)	√	
10.	2.75	279	783.0702	481.0604; 300.9982	Bis-HHDP-hexoside (pedunculagin isomer)	√	√
11.	3.30	235	633.0581	481.9926/300.9982	HHDP-galloyl-hexoside	√	
12.	3.65	326	367.0090	191.0128	Methyl-caffeoyl-quinate	√	
13.	3.66	274	783.0645	481.0613; 300.9984	Bis-HHDP-hexoside (pedunculagin isomer)		√
14.	4.39	280	935.0596	783.0650; 633.0740	Galloyl-bis-HHDP-hexoside	√	√
15.	4.5	235/320	785.0615	633.0641; 300.9984	HHDP-digalloyl-hexoside	√	√
16.	4.82	325	633.0550	481.0470; 300.9982	HHDP-galloyl-hexoside	√	√
17.	4.93	230	933.0406	915.0529; 633.0581; 450.9908; 301.0070	Castalagin/vescalagin		√
18.	5.07	224	936.0582	300.9964	Galloyl-bis-HHDP-hexoside		√
19.	5.32	279	783.0427	481.0117; 300.9964	Bis-HHDP-hexoside (pedunculagin isomer)	√	√
20.	5.51	235	933.0645	450.9958; 301.0000	Castalagin/vescalagin		√
21.	5.68	246	1869.1495	934.0719; 633.0584; 468.9935; 300.9985	Dimer of galloyl-bis-HHDP-glucose (sanguiin H-6)	√	√
22.	5.85	230	933.0707	633.0404; 300.9991	Castalagin/vescalagin		√
23.	6.17	252/364	433.0420	301.0354	Ellagic acid pentoside	√	√
24.	6.5	255/365	300.9999	285.0425; 257.0208; 229.0137	Ellagic acid	√	√
25.	6.87	225	935.0639	783.0604; 300.9987	Galloyl-bis-HHDP-hexoside		√
26.	7.12	224	935.0799	783.0650; 300.9981	Galloyl-bis-HHDP-hexoside		√
27.	7.56	350	447.0560	315.0143	Isorhamnetin pentoside	√	√
28.	7.75	254/362	461.0121	300.9984	Ellagic acid hexoside	√	
29.	8.31	340	315.0133		Isorhamnetin	√	√

^a^ Abbreviations: Rt, retention time; HHDP, hexahydroxydiphenyl; ^b^ Experimental data.

**Table 2 molecules-27-02804-t002:** Phenolic composition of different extracts of *Alder* acorns (mg/100 g of dry matter) *.

Peak	Compound	Water	Ethanol
	*Ellagitannins*		
1.	HHDP-hexoside (1-galloyl-2,3,hexahydroxydiphenoyl-α-glucose)	43.43 ± 0.5 k	0.00 ± 0.0 l
3.	HHDP-hexoside	143.35 ± 4.5 d	0.00 ± 0.0 l
4.	Trisgalloyl hexoside	68.04 ± 1.2 i	0.00 ± 0.0 l
5.	Methoxylated castalagin/vescalagin	97.50 ± 2.1 g	0.00 ± 0.0 l
6.	HHDP-digalloyl-hexoside	173.31 ± 3.7 c	0.00 ± 0.0 l
7.	Bis-HHDP-hexoside (pedunculagin isomer)	242.52 ± 5.5 a	0.00 ± 0.0 l
8.	Digalloyl-glucose	82.31 ± 1.1 h	0.00 ± 0.0 l
9.	Bis-HHDP-hexoside (pedunculagin isomer)	103.60 ± 2.3 f	0.00 ± 0.0 l
10.	Bis-HHDP-hexoside (pedunculagin isomer)	225.91 ± 3.4 a	199.84 ± 2.4 b
11.	HHDP-galloyl-hexoside	72.26 ± 1.0 i	0.00 ± 0.0 l
13.	Bis-HHDP-hexoside (pedunculagin isomer)	0.00 ± 0.0 l	173.24 ± 1.9 c
14.	Galloyl-bis-HHDP-hexoside	49.69 ± 1.2 k	232.35 ± 2.2 a
15.	HHDP-digalloyl-hexoside	59.48 ± 1.1 j	96.43 ± 0.8 g
16.	HHDP-galloyl-hexoside	63.44 ± 1.6 j	105.95 ± 1.0 f
17.	Castalagin/vescalagin	0.00 ± 0.0 l	102.70 ± 1.0 f
18.	Galloyl-bis-HHDP-hexoside	0.00 ± 0.0 l	210.04 ± 1.6 b
19.	Bis-HHDP-hexoside (pedunculagin isomer)	108.28 ± 2.3 f	208.00 ± 1.3 b
20.	Castalagin/vescalagin	0.00 ± 0.0 l	215.44 ± 1.5 b
21.	Dimer of galloyl-bis-HHDP-glucose (sanguiin H-6)	59.99 ± 0.9 j	206.01 ± 1.2 b
22.	Castalagin/vescalagin	0.00 ± 0.0 l	183.46 ± 1.0 c
25.	Galloyl-bis-HHDP-hexoside	0.00 ± 0.0 l	130.60 ± 0.9 e
26.	Galloyl-bis-HHDP-hexoside	0.00 ± 0.0 l	107.83 ± 0.8 f
	Sum	1593.13	2171.90
	*Ellagic acid derivatives*		
23.	Ellagic acid pentoside	247.78 ± 10.2 d	409.99 ± 15.6 c
24.	Ellagic acid	675.37 ± 19.7 a	582.56 ± 13.3 b
28.	Ellagic acid hexoside	29.13 ± 0.2 e	0.00 ± 0.0 f
	Sum	952.28	992.54
	*Phenolic acids*		
2.	Ferulic acid hexoside	68.48 ± 0.3 b	0.00 ± 0.0 c
12.	Methyl-caffeoyl-quinate	195.09 ± 0.9 a	0.00 ± 0.0 c
	Sum	263.57	0.00
	*Flavonols*		
27.	Isorhamnetin pentoside	19.80 ± 0.8 b	53.25 ± 1.3 a
29.	Isorhamnetin	21.16 ± 0.7 b	59.03 ± 1.5 a
	Sum	40.96	112.28
	Sum of phenolic compounds	2849.93	3276.72

* Values are means ± standard deviation. *n* = 3. Mean values within a row with different letters (a–l) are significantly different at *p* < 0.05; Amounts of phenolic acids, flavonols, ellagitannins and ellagic acid derivatives, were converted into ferulic acid (ferulic acid derivatives), caffeic acid (methyl-caffeoyl-quinate), isorhamnetin 3-*O*-glucoside (isorhamnetin derivatives) and ellagic acid (ellagitannins and ellagic acid derivatives).

**Table 3 molecules-27-02804-t003:** Effect of plant extracts on growth of tumor cells lines.

Dilution of Extract Stock Solution	The Percentage of Viability			
	A431 Cells		A549 Cells	
	Ethanol Extract	Water Extract	Ethanol Extract	Water Extract
1:400	51 ± 2 ^f^	37 ± 2 ^f^	77 ± 5 ^f^	58 ± 1 ^f^
1:800	61 ± 2 ^f^	40 ± 1 ^f^	77 ± 3 ^f^	62 ± 1 ^f^
1:1600	68 ± 6 ^f^	38 ± 1 ^f^	80 ± 1 ^f^	67 ± 1 ^f^
1:3200	77 ± 2 ^f^	55 ± 3 ^f^	95 ± 1	71 ± 2 ^f^

Cells were incubated with indicated extract dilutions for 72 h, followed by determination of cells viability with MTT colorimetric method. The results are shown as percentage of cells’ viability versus control (cultures containing appropriate dilutions of the solvent; H_2_O or Ethanol) (mean values from three independent experiments). The percentage of viability was calculated in accordance with equation: % cell viability = [A extract/A control] × 100; where A extract—is the absorbance of the sample exposed to the extract of plants; A control—is the absorbance of the control sample—cell without extract exposition—100% cell viability. ^f^ for *p* < 0.05, versus Control probes.

**Table 4 molecules-27-02804-t004:** Antibacterial and antifungal effects of water and ethanol extracts of *Alder* acorns.

	Bacterial or Fungal Growth Inhibition Zone (mm)					
Strains	Water Extract Dilution			Ethanol Extract Dilution		
	1:1	1:5	1:10	1:20	1:40	1:80
*Escherichia coli* PCM 2057	6 mm (clear zone)	6 mm	3 mm	6 mm (clear zone)	0	0
	(21 mm opaque)					
*Pseudomonas aeruginosa* PCM 2058	0	0	0	0	0	0
*Pseudomonas aeruginosa* PCM 499	0	0	0	0	0	0
*Proteus mirabilis* PCM 2958	0	0	0	0	0	0
*Citrobacter freundii* PCM 2959	0	0	0	0	0	0
*Staphylococcus aureus* PCM 2054	12 mm clear zone	11 mm (opaque)	4 mm (opaque)	5 mm (clear zone)	5 mm (opaque)	0
*Streptococcus mutans* ATCC 25175	0	0	0	0	0	0
*Bacillus subtilis* PCM 2021	6 mm clear zone	11 mm (opaque)	0	5 mm (clear zone)	4 mm (clear zone)	0
	(21 mm opaque)					
*Candida albicans* ATCC 90028	6 mm (opaque)	0	0	4 mm opaque	0	0
*Candida glabrata* PCM 2703	12 mm (opaque)	11 mm (opaque)	4 mm (opaque)	6 mm opaque	5 mm (opaque)	0
			Ethanol dilution equivalent	0	0	0

**Table 5 molecules-27-02804-t005:** Effect of water extract of *Alder* acorns on bacteria number.

Bacteria	CFU/mL/Time		
	0 h	24 h	
		Control	Extract Alder Acorns
*Staphylococcus aureus* PCM 2054	1.0 × 10^7^	7.5 × 10^7^	7.0 × 10^5^ *
*Escherichia coli* PCM 2057	2.0 × 10^8^	6.0 × 10^8^	4.6 × 10^7^ *

* for *p* < 0.05, versus Control probes (24 h).

## Data Availability

Data are contained within the article.

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
