# Peer review of "Chemical Characteristics of Ethanol and Water Extracts of Black Alder (Alnus glutinosa L.) Acorns and Their Antibacterial, Anti-Fungal and Antitumor Properties"

_molecules, 2022, doi:10.3390/molecules27092804_

Round 1
Reviewer 1 Report
The innovation and main objectives of the work are well described. The discussion of the data is in accordance with the proposal and is highly relevant.
I would just like to suggest to authors that they add a graphic abstract of the work routes (in the methodology section), so that it is more attractive to the readers.
I recommend accepting this work in its present form after this addition.
Author Response
Thank you for the review and all comments and tips. Here are our answers:
- I would just like to suggest to authors that they add a graphic abstract of the work routes (in the methodology section), so that it is more attractive to the readers.
Reply: Graphic abstract is added as an separate file

Reviewer 2 Report
This work is focused on determination of phytochemical profiles of black alder acorns extracts obtained by maceration using ethanol and water as solvents. The major contribution of the presented work is seen in the isolation of a wide spectrum of phenolics from black alder extracts and the evaluation of their anti-cancer and antimicrobial effects. The results provided basic approach for treating black alder acorns enriched in different phytochemicals to enhance its bio-accessibility. Accordingly, major revision should be made before it is published in the Journal. The comments are as follows.
- The authors are encouraged to revise the abstract and improve it by inserting some results.
- The introduction section is too long; in my opinion should be resumed. Indeed, the paragraphs from line 45 to 92 are too descriptive. Thus, it should be rewritten highlighting the aims of the research.
- The particle size and moisture content of black alder acorns should be provided.
- The samples of plants were extracted using 70% ethanol via maceration in darkness at room temperature or being held at the boiling point for 60 min at 100°C. High temperatures may lead the degradation and loss of target compounds. Please discuss a little bit on this concern.
- Incorporate statistical analysis of the results (comparison between extraction methods in terms of contents of compounds, anti-cancer and antimicrobial effects).
- In my opinion, a discussion about the relation between the compounds identified in the extract and the different bioactivities exerted should be included in order to be considered for publication. Respect to the antimicrobial assay, the authors should justify more the different activity of the extracts against the tested microorganisms.
Author Response
Thank you for the review and all comments and tips. Here are our answers:
The authors are encouraged to revise the abstract and improve it by inserting some results.
Reply: Abstract has been revised
The introduction section is too long; in my opinion should be resumed. Indeed, the paragraphs from line 45 to 92 are too descriptive. Thus, it should be rewritten highlighting the aims of the research.
Reply: Introduction has been corrected
The particle size and moisture content of black alder acorns should be provided.
Reply: Information added to text: The average size of the cones 1.5 cm, before extraction, they were homogenized in a laboratory mill. Particle size wasn’t measured – it was homogenous powder. The cones' moisture was at the level of 11.17%.
The samples of plants were extracted using 70% ethanol via maceration in darkness at room temperature or being held at the boiling point for 60 min at 100°C. High temperatures may lead the degradation and loss of target compounds. Please discuss a little bit on this concern.
Reply: Discussion is placed in lines 247-279 (lines 196-227 after you accept all changes)
Incorporate statistical analysis of the results (comparison between extraction methods in terms of contents of compounds, anti-cancer and antimicrobial effects).
Reply: This is now corrected
In my opinion, a discussion about the relation between the compounds identified in the extract and the different bioactivities exerted should be included in order to be considered for publication. Respect to the antimicrobial assay, the authors should justify more the different activity of the extracts against the tested microorganisms.
Reply: Discussion has been added

Reviewer 3 Report
Dear authors,
This is a potential submission towards Molecules. The results are novel and very interesting. My suggestions and comments to improve the manuscript are included in the attached file. Please note the entire manuscript must be revised carefully for typos, English, and scientific names must be checked.
With kind regards.

Author Response
Thank you for the review and all comments and tips. Here are our answers:
Candida: This is a genus and not a species. Please revise carefully. You have to homogenize by adding bacteria names
Reply: This is now corrected
Line 21: Please be precise: I suggest the use of A. glutinosa extracts
Reply: Has been corrected
Line 33: Scientific names must be revised across the entire manuscript
Reply: This is now corrected
Line 38: First letter of genus and species
Reply: Has been corrected
Line 49: Please change to "able to increase"
Reply: Has been corrected
Line 87: Please rephrase
Reply: Has been rephrased
Lines 94,96: A. glutinosa
Reply: Has been corrected
Table 1: Please check for typos
Reply: Has been checked
Line 206: Please change to "In the aqueous extract, "
Reply: Has been corrected
Line 228: I suggest to deepen discussion section in light of these recent references (10.33263/BRIAC126.84418452; 10.33263/BRIAC114.1144011457)
Reply: Discussion has been added
Table 3: Please note that results must be statistically analyzed
Reply: This is now corrected
Lines 298-300: Please justify the use of such strains
Reply: This is now corrected
Table 4: Statistical analyses must be done
Reply: This is now corrected
Lines 347-351: Please see below an interesting reference to compare with and explain the observed differences (https://doi.org/10.33263/BRIAC112.93589371)
Reply: The work indicated by the reviewer was a physicochemical analysis, no reference to biological activity
Line 375: A brief description of climatic conditions of sampling site must be added. Also, phenology at which plant has harvested must be provided
Reply: Alder fruits were collected from wild trees, so there is no detailed climatic research. The only thing that can be written is that: The climate in Kalisz is warm and temperate. There is significant rainfall throughout the year in Kalisz. Even in the driest month there is a lot of rainfall. The climate here is classified as Cfb by the Köppen-Geiger system. The average temperature in Kalisz is 9.6 ° C. The average annual rainfall in this area is 666 mm.
The period of phenology is early fall. The harvest date is given and this seems to be sufficient for this type of research purpose.
Line 460: Please revise for typo
Reply: This is now corrected
Conclusion: Conclusions section must be rewritten to highlight the main conclusions and not stating some results reported in the manuscript
Reply: Conclusion has been corrected

Reviewer 4 Report
Nawirska-Olszańska and colleagues present a study on the chemical composition, antibacterial, antifungal, and antitumor properties of ethanolic and acqueous extracts of Alnus glutinosa L.
The study was well-conceived and executed.
Only minor revisions are required, and the authors are encouraged to consider the following:
- There are a grammatical and typographical issues that should be fixed.
- To cite research by a research group is suitable to name only the first author followed by “et al.” (lines 33-34; 222-223; 349; 353; 357-358)
- “botulin” in line 37 must be changed in “betulin”
- In Table 2, there are many letters in water and ethanol content but there is not the explanation of their significance.
- In the paragraph 2.4. Antimicrobial activity, in lines 316-318 are referred the activity of ethanol extract. It should be introduced in the text which strain of microorganism and fungi are inhibited, similar to water extract. The only reference to Table 4, should be avoided.
- In lines 340-341 should be corrected the exponential numbers.
Author Response
Thank you for the review and all comments and tips. Here are our answers:
- There are a grammatical and typographical issues that should be fixed.
Reply: Text was checked by native-speaker
- To cite research by a research group is suitable to name only the first author followed by “et al.” (lines 33-34; 222-223; 349; 353; 357-358)
Reply: Has been corrected
- “botulin” in line 37 must be changed in “betulin”
Reply: Has been corrected
- In Table 2, there are many letters in water and ethanol content but there is not the explanation of their significance.
Reply: Information has been added
In the paragraph 2.4. Antimicrobial activity, in lines 316-318 are referred the activity of ethanol extract. It should be introduced in the text which strain of microorganism and fungi are inhibited, similar to water extract. The only reference to Table 4, should be avoided.
Reply: This is now corrected
In lines 340-341 should be corrected the exponential numbers.
Reply: This is now corrected

Round 2
Reviewer 2 Report
The authors accepted all the reviewer's comments and improved the quality of the paper.
Reviewer 3 Report
Dear Authors,
The submission has been improved and therefore I suggest its publication in Molecules.
Regards.